# Cancer Mortality Trend in Central Italy: Focus on A “Low Rate of Land Use” Area from 1982 to 2011

**DOI:** 10.3390/ijerph16040628

**Published:** 2019-02-21

**Authors:** Francesca Santilli, Stefano Martellucci, Jennifer Di Pasquale, Cecilia Mei, Fabrizio Liberati, Carmela Protano, Matteo Vitali, Maurizio Sorice, Vincenzo Mattei

**Affiliations:** 1Laboratory of Experimental Medicine and Environmental Pathology, Rieti University Hub “Sabina Universitas”, 02100 Rieti, Italy; f.santilli@sabinauniversitas.it (F.S.); stefano.martellucci@uniroma1.it (S.M.); jenndip1102@gmail.com (J.D.P.); c.mei@sabinauniversitas.it (C.M.); vincenzo.mattei@uniroma1.it (V.M.); 2Department of Experimental Medicine, Sapienza University of Rome, 00185 Rome, Italy; maurizio.sorice@uniroma1.it; 3UOC Anatomic Pathology, San Camillo De Lellis Hospital, 02100 Rieti, Italy; f.liberati@asl.rieti.it; 4Department of Public Health and Infectious Diseases, Sapienza University of Rome, 00185 Rome, Italy; carmela.protano@uniroma1.it

**Keywords:** cancer, mortality, epidemiological investigation, Italy

## Abstract

The aim of the present study was to estimate total cancer mortality trends from 1982 to 2011 in a “low rate of land use” province of the Latium region (Rieti, central Italy) characterized by a low degree of urbanization, a high prevalence of elderly, and a low number of births. Mortality data of the studied period, provided by the Italian National Institute of Statistics, were used for calculating standardized cancer mortality rates. Trends in mortality were analyzed using Joinpoint regression analysis. Results showed that total standardized cancer mortality rates decreased in the monitored area over the study period. A comparison with other provinces of the same region evidenced that the studied province presented the lowest cancer mortality. The three systems/apparatuses affected by cancer that mainly influenced cancer mortality in the monitored province were the trachea-bronchus-lung, colorectal-anus, and stomach. These findings could be attributed to the implement of preventive initiatives performed in the early 2000s, to healthier environmental scenario, and to lower levels of carcinogenic pollutants in air, water, and soil matrices. Thus, our results indicate that the studied area could be considered a “healthy” benchmark for studies in oncological diseases.

## 1. Introduction

Cancer remains one of the most common causes of death worldwide. Indeed, a recent systematic analysis for the global burden of diseases highlighted that cancer was the second leading cause of death. Globally, it is estimated that there were 18.1 million new cancer cases and 9.6 million cancer deaths in 2018 [1]. In Italy, the number of deaths attributed to cancer in 2014 was 177,301 and, likewise in other developed countries, cancer was the second cause of death (29% of all deaths) after cardiovascular diseases (37%) [2]. In Italy, in 2017, over 369,000 new cases of cancer were diagnosed (54% in men and 46% in women) [2].

Over the past several decades, scientific evidence has demonstrated the relevance of several risk factors, such as environmental pollutants [3], genomic alteration [4], infectious diseases [1], lifestyles and high body mass index, poor intake of fruit and vegetables, sedentarity, tobacco and alcohol consumption, and aging [5]. In particular, aging seems to be a key factor in the development of cancer, and the incidence of most cancers significantly increases with age. This relationship is linked to the longer duration of exposure to risk factors and to the decrease of defending ability and repairing mechanisms [6]. Additionally, specific age-related epigenetic changes and the so-called “Hayflick limit” (typical aging factors such as telomere loss, genomic instability, oxidative stress, and the accumulation of DNA damage) have been recognized as key linkages between aging and cancer [7,8]. Consequently, demographic changes are a relevant predictor of the number of diagnosed cancers. In view of the increase in life expectancy, the total number of new cases of cancer is expected to grow over time. However, while the incidence of cancer has increased in most countries, the mortality has decreased. Indeed, globally, cancer mortality rates have declined steadily in higher as well as in lower income regions and in both sexes [9]. In this scenario, Italy is one of the worst countries, together with France, Australia, and Japan, since cancer incidence has highly increased in the last ten years [10]. This phenomenon could be related to the structure of the population. In fact, life expectancy at birth of the Italian population has significantly grown over the years [11]. However, in Italy, total cancer mortality has declined since the 1980s [12], similarly to most European countries and the United States [13,14]; this reduction can be attributed to several factors, such as the effectiveness of prevention strategies and, in particular, to the implementation of national screening programs and to the improvement in the use of multidisciplinary and integrated therapies [15]. Unfortunately, a complete and appropriate evaluation of the temporal trends of Italian cancer deaths is not possible to date, because a systematic data collection on a national basis does not exist. On the other hand, the World Health Organization (WHO) strongly recommends the creation of a baseline for monitoring cancer trends in order to contrast the global cancer epidemic and to evaluate the progress of countries in addressing this epidemic. Thus, it is essential to produce evidence and to trace epidemiological profiles of cancer through ad hoc studies. In our previous studies [16,17], we reported preliminary data on cancer mortality trends of the five provinces of the Latium region. The present study aimed to estimate total cancer mortality trends of the thirty-year period between 1982 and 2011 with a particular focus on Rieti, a “low rate of land use” province of the Latium region (central Italy) characterized by a low degree of urbanization, a high prevalence of elderly [18], and a low number of births [19].

## 2. Materials and Methods

### 2.1. Study Area

The main characteristics in term of population density (as a proxy indicator of urbanization degree), demographic aging, and some environmental features of the study area are shown in Table 1 and Table 2. The same data are reported also for the other Latium provinces (Rome, Latina, Frosinone, and Viterbo). In total, the Latium region presents a land area of 17,203 km² (5.7% of the whole Italian territory). In particular, Table 1 reports the population density, calculated by dividing the total population (as the number of people) by the land area covered by that population, and the aging index, calculated by dividing the number of elderly persons (aged 65 and over) by the number of young persons (from 0 to 14 years old) × 100 at the years 1981, 1991, 2001, and 2011 (Italian censuses).

Table 1 evidenced that the monitored area presents, respectively, a lower population density and a higher ageing index with respect to both the other provinces of the Latium region and to Italy in general.

Moreover, Table 2 shows some of the main environmental characteristics of the Latium provinces.

As reported in Table 2, the study area presents low levels of well-known carcinogenic air, water, and soil pollutants. Additionally, regional data evidenced that the study area is characterized by low industrial activities, low anthropogenic impact, low vehicular traffic [20], and low municipal solid waste production [21].

### 2.2. Data Collection

The present study was performed using data on cancer mortality, provided by the Italian National Institute of Statistics (ISTAT), for Rieti province and the Latium region for the thirty-year period from 1982 to 2011. In particular, ISTAT furnished data stratified by sex (male and female), five-year age groups (0–4; 5–9; 10–14; etc.), and type of cancer coded based on the International Statistical Classification of Diseases, Injuries and Causes of Death (ICD). Specifically, data were coded according to the ninth revision for the years from 1982 to 1991 (ICD-9, codes 140–239) and the tenth revision from 1992 to 2011 (ICD-10, codes C00–D48) [22].

### 2.3. Data Processing and Statistical Elaboration

Data collected for each considered year were processed to obtain the number of deaths (total number and according to gender). It should be noted that the colorectal and anus cancers were classified differently in the two revisions of ICD. For this reason, we decided to re-elaborate the data in order to have a single anatomic site for this type of cancer (colorectal-anus cancer) for all the investigated years. Additionally, we considered the resident population of Rieti province and the Latium region, divided by gender and age groups, as the arithmetic mean of the number of inhabitants until 1 January and the number of inhabitants until 31 December of each year. Using these data, we calculated total and gender standardized cancer mortality rates by the direct method as the ratio between total expected deaths and total reference population (2001 Italian census) ×10,000 inhabitants for each year. Total standardized mortality rates were also calculated for the three most common cancer anatomic sites: trachea-bronchus-lung, colorectal-anus, and stomach. Moreover, we compared total and gender percentage variation in the standardized mortality rate for decades (first decade vs. third decade). Furthermore, we also compared total and specific standardized mortality rates for the three common anatomic sites for Rieti province and Latium Region.

MS-Excel software (Microsoft Office 365 Suite, Microsoft Corporation, Redmond, WA, USA) was used to calculate standardized rates by sex and age groups.

The statistical significance of the variation in mortality trends and the annual changes in standardized rates were analyzed using Joinpoint software version 4.6.0.0; this software was developed by the US National Cancer Institute for the trend analyses of data from the Surveillance Epidemiology and End Results Program [23]. In particular, the program can be used for assessing the statistical significance in the change over time in linear slope of the trend through a Monte Carlo permutation method. Additionally, the software estimates an annual percent change (EAPC) computed for each trend using a generalized linear model assuming a Poisson distribution. Significance of the change in standardized rates were evaluated with *p*-values, using a significance level of 0.05.

## 3. Results

Figure 1 shows the trends of cancer mortality (standardized rates per 10,000 inhabitants) in the monitored area from 1982 to 2011, both for the entire study population and for the subpopulation of females and males.

The results presented in Figure 1 show a declining trend in mortality rates, both for males and females. The statistical significance of the variation in mortality trends and the annual changes in standardized rates were evaluated by the use of the Joinpoint regression analyses. The Joinpoint trend graphs for standardized cancer mortality rates and annual percent change (APC) with 95% Confidence Interval and the related *p*-values, according to gender, are reported in Figure 2.

The results of the Joinpoint analysis confirm a progressively, even if not statistically significant, decrease of the standardized mortality rates for both males and females over the monitored period (males: APC = −0.1, 95% CI −0.3–0.2, *p*-value > 0.5 for trend; females: average (A) APC = −0.5, 95% CI −1.2–0.2, *p*-value = 0.2 for trend).

The analysis of the standardized cancer mortality rates in relation to the types of cancer highlighted that trachea-bronchus-lung cancer causes the greatest number of deaths in the monitored area, followed by colorectal-anus, stomach, breast, prostate, liver, pancreas, and leukemia cancers. Given this preliminary result, we focused the attention on the three deadliest cancers.

Figure 3 reports the standardized mortality rates (total population and grouped according to gender) for the three deadliest cancers in the monitored area for all the investigated years.

As shown in Figure 3a, the standardized trachea-bronchus-lung cancer mortality rate increased from 3.5 in 1982 to 4.4 in 2011 ×10,000 inhabitants, with a 7.5% increase from the first to the last year under investigation. With regard to the colorectal-anus cancer, the trend of the total standardized mortality rate showed a substantial invariance from 2.3 in 1982 to 2.5 in 2011 ×10,000 inhabitants. With respect to stomach cancer, the trend of the total standardized mortality rate showed a decrease, from 2.8 in 1982 to 1.8 in 2011 ×10,000 inhabitants.

We also independently evaluated data related to the subpopulations of males and females; this analysis highlighted differences in the mortality trend for the three types of cancer considered in the two subpopulations (Figure 3b,c). First of all, the gender analysis evidenced that the standardized trachea-bronchus-lung cancer mortality rate was about seven times higher in the male subpopulation than in the female subpopulation at the start of the investigated period. Over the years, however, this difference has been slightly reduced due to the increasing number of deaths for trachea-bronchus-lung cancer among women. Indeed, the percentage variation in the standardized trachea-bronchus-lung cancer between the first and the third decade revealed a substantial invariance in males (<1%) and an inverse trend in females, with an increase of 80.0%. Furthermore, gender trend evaluation also indicated a different behavior for colorectal-anus cancer. In fact, the comparison of the percentage variation in the standardized colorectal-anus mortality rate evidenced that the standardized mortality rate for males has increased 3.1%, while in females it has decreased 8.3% (first decade vs. third decade. Finally, the decreasing trend of stomach cancer in the total population was confirmed both for the male and female groups.

Table 3 reports the cancer mortality trends (mean value for each five-year interval) for the monitored area and the related region (Rieti and Latium region, respectively).

The comparison of the total mean of the standardized mortality rate between Rieti province and the Latium region showed that the monitored area had a lower standardized cancer mortality rate, with a thirty-year mean of 25.3 vs. 29.9 for the Latium region.

Figure 4 reports a comparison of the thirty-year mean for the standardized mortality rates of the deadliest cancers.

Figure 4 shows that Rieti had lower mortality rates than those of the Latium region for both trachea-bronchus-lung cancer (4.1 vs. 6.3) and colorectal-anus cancer (2.7 vs. 3.3). In contrast, the mortality trend was similar for stomach cancer (2.6 vs. 2.4). This trend was confirmed by gender analysis, with the same differences observed separately for males and females.

## 4. Discussion

Presently, cancer is one of the most important diseases in term of incidence and mortality worldwide; thus, surveillance programs for tracing the epidemiological profiles of cancer are essential in order to inform appropriate choices in terms of public health strategies. Despite this, in many countries, including Italy, there are no routine surveillance programs and no benchmark areas have been established to carry out appropriate comparisons; this gap can be partially filled by ad hoc studies describing epidemiological profiles of cancer in specific areas. We studied the cancer mortality rate in a “low rate of land use” area for a long period of time (1982–2011).

The first relevant finding of the present study is related to the general decrease in the cancer mortality rates in the monitored area, both for males and females. This result is in line with those reported for global mortality rates [24] and for specific countries, such as European countries [25], the United States [26], Canada [27], and Japan [28]. The consistent decrease in cancer mortality rates can be attributed to the effectiveness of prevention interventions in reducing the prevalence of exposure to main risk factors, to screening and early detection campaigns, and to the improvement of valid and appropriate treatments.

Another relevant finding is related to the mortality rates of the deadliest cancers: in the investigated period we observed that the types of cancer responsible for the greatest number of deaths in the monitored area were trachea-bronchus-lung, colorectal-anus, and stomach cancers, with some notable gender differences. Indeed, males’ mortality rates were in line with those found for the total population, while females’ mortality rates presented some peculiarities: breast cancer was associated with the highest mortality rates (data not shown), followed by colorectal-anus and stomach cancers. However, the analysis of the standardized mortality rates from 1982 to 2011 demonstrated that, even if trachea-bronchus-lung tumors were not the leading cause of cancer-related deaths in women, their mortality rates increased substantially in the female subpopulation over the years. This phenomenon is comparable to the trends observed in other European countries, where females’ mortality rates for lung cancer have shown significant increasing trends [29,30,31].

The second type of cancer most affecting the cancer-related mortality in the monitored area was colorectal-anus cancers, with no significant changes in the total standardized mortality rate in the studied period. However, rates for this type of cancer presented some gender differences: there was a 3.1% increase in males and a reverse trend in females, with a decrease of 8.3%. These results were confirmed by a recent study carried out in Italy on the same type of cancer that showed a slow but steady decrease in total mortality (1999 to 2016) [6]. Likely, these findings could be related to diagnostic improvements and the implementation of screening programs [32,33,34,35] that occurred from the early 2000s and onwards.

The third type of cancer that was responsible for the highest mortality rates in Rieti province was stomach cancer. The analysis of total standardized mortality rates over the thirty-year period showed a steady and progressive decline from 2.8 in 1982 to 1.8 in 2011 per 10,000 inhabitants, confirmed in both sexes, with values of −35.7% for males and −50.0% for females. These results agree with those observed in Italy (1999 to 2016), as a marked decrease both in incidence and mortality has been observed for this type of cancer, which could be partly attributed to better food preservation practices, the improvement of diagnostic techniques, and the more appropriate diagnosis and treatment of *Helicobacter pylori* infection, a known risk factor for gastric cancer [36,37,38,39].

The comparison between standardized mortality rates found in Rieti province and the Latium region showed that Rieti had the lowest values throughout the observed period. The present study, performed with an ecological approach, cannot prove any causal correlation between an exposure and an event. Thus, it is not possible to provide an explanation of the differences observed between the studied provinces. However, it is well known that the main modifiable risk factors associated with cancer are behavioral and environmental. We do not have specific information on the prevalence of behavioral risk factors in the studied area, although environmental data are made available by the monitoring campaigns performed routinely by the Regional Agency for Environmental Protection and Prevention of the Latium Region. They show that some well-known carcinogenic pollutants, including benzene, PM_10_, arsenic, and radon, are present in Rieti province at concentrations lower than those found in the other provinces of the Latium region [16]. In addition, the Rieti area is characterized by a low population density, many green areas, and a very low anthropogenic impact in terms of environmental impacts (i.e., industrial plants, landfills, incinerators, etc.). Moreover, the improvement in the standardized rate observed since the year 2000 in Rieti province could be related to the strengthening of the prevention campaigns and the implementation of new therapeutic protocols following the opening of oncology (1997) and radiotherapy (2004) units in the local general hospital, as well as the creation of an interdisciplinary team for oncology care (2007). The present study has some limitations. Firstly, the ecological approach does not permit any causal correlation to be established between an exposure and an event. Additionally, data used for the calculation of mortality rates were not collected by authors but obtained from ISTAT, which elaborated data reported on the death certificates of all Italian population. These data were not grouped according to age nor was age of death reported for each individual. Thus, we were not able to calculate cancer mortality in those monitored areas after age adjustment, as elaborated by other researchers [40,41]. Finally, during the studied period, the ICD classification changed and thus we were forced to use two different classification systems (ICD-9 and ICD-10). However, given the lack of a systematic collection of data on cancer mortality in the studied area, this analysis provides a snapshot of the trends in cancer mortality over thirty years.

## 5. Conclusions

The present study shows both a decreasing temporal trend and a lower mortality rate for cancer in Rieti province when compared to the other Latium provinces. These findings could be attributed to the implementation of preventive initiatives performed in the early of 2000s, to healthier environmental conditions, and to lower levels of carcinogenic pollutants in air, water, and soil matrices. Thus, even if we cannot provide any casual correlation between exposures to pollutants and cancer mortality trend, our results indicate that the studied area could be considered as a “healthy” benchmark for studies on oncological diseases.

## Figures and Tables

**Figure 1 ijerph-16-00628-f001:**
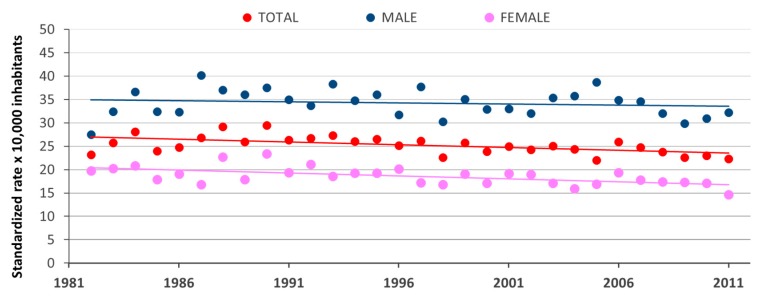
Trend of cancer mortality (standardized rates per 10,000 inhabitants) in the monitored area from 1982 to 2011 (red line for total population, pink line for females, blue line for males).

**Figure 2 ijerph-16-00628-f002:**
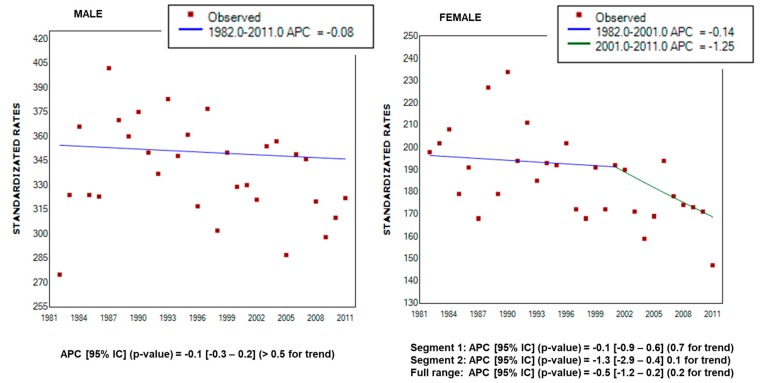
Joinpoint trend graphs for standardized cancer mortality rates for males and females in Rieti province from 1982 to 2011. APC: annual percent change.

**Figure 3 ijerph-16-00628-f003:**
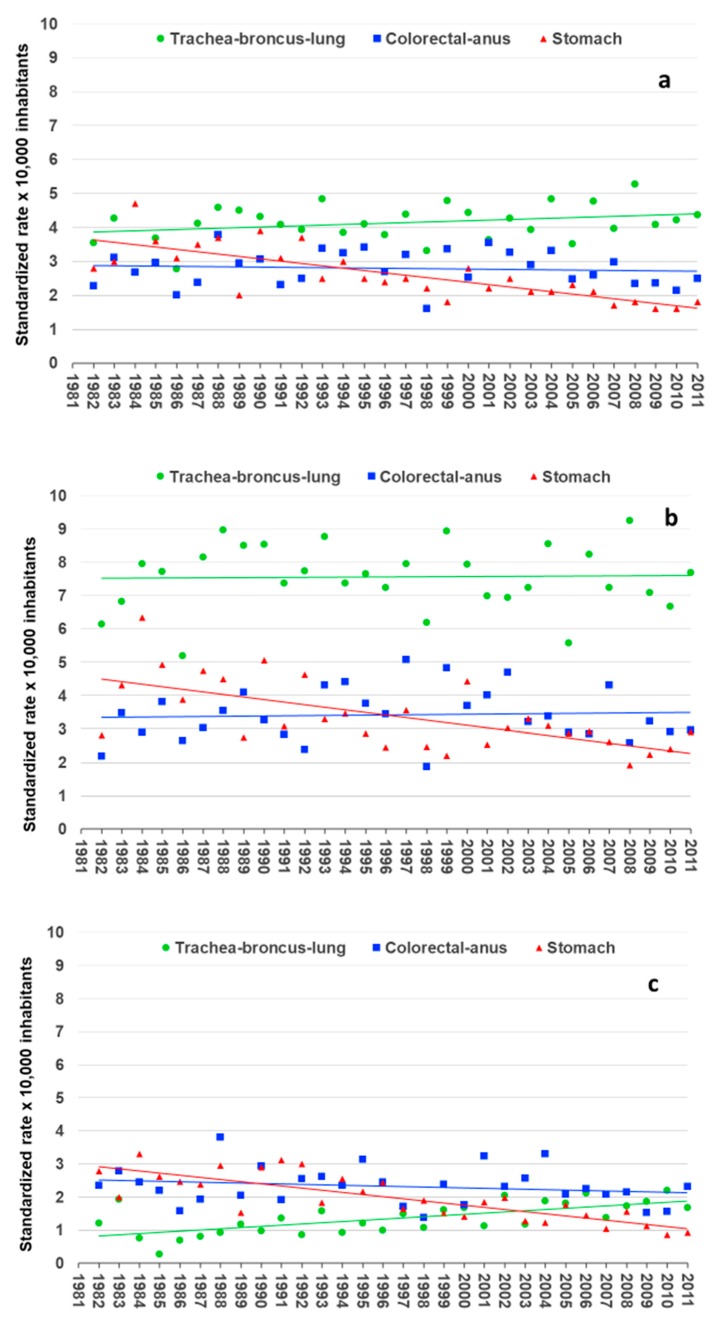
Trend of cancer mortality (standardized rates per 10,000 inhabitants) for the deadliest cancers in the monitored area from 1982 to 2011: (**a**) total population; (**b**) male subpopulation; (**c**) female subpopulation.

**Figure 4 ijerph-16-00628-f004:**
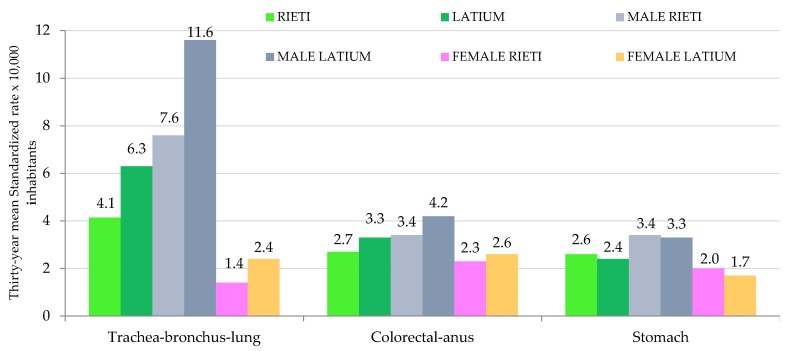
Thirty-year mean standardized rates of the deadliest cancers in the monitored area (Rieti) and the Latium region (both for total population, male and female subpopulations).

**Table 1 ijerph-16-00628-t001:** Population density and aging index of five provinces of the Latium region and Italy (1981–2011 Italian census).

Year	Population Density (People per sq km of Land Area)						Ageing Index (%)					
	Rieti	Rome	Latina	Frosinone	Viterbo	Italy	Rieti	Rome	Latina	Frosinone	Viterbo	Italy
1981	51.9	689.1	192.4	141.8	74.3	187.2	97.4	53.2	38.9	58.8	75.6	61.7
1991	52.7	701.3	211.5	148.1	77.6	187.9	139.7	94.9	62.9	82.5	114.4	92.5
2001	53.4	690.8	217.9	149.2	79.8	188.7	172.3	130.9	99.6	125.5	163.6	129.3
2011	56.4	743.4	240.8	151.8	86.4	196.7	188.9	141.5	124.9	152.5	172.8	145.7

**Table 2 ijerph-16-00628-t002:** Annual mean concentrations of some environmental pollutants monitored in the Latium provinces (Regional Agency for Environmental Protection and Prevention of the Latium Region).

Pollutants	Rieti	Rome	Latina	Frosinone	Viterbo
Airborne benzene (mg m^−3^) ^1^	2.2	3.3	2.6	4.2	2.2
Airborne PM_10_ (mg m^−3^) ^2^	21.8	32.2	28.3	36.4	24.1
Airborne PM_2.5_ (mg m^−3^) ^3^	12.5	17.2	16.8	21.2	11.6
Arsenic in drinking water (mg L^−1^) ^4^	0.6	5.0	7.6	7.2	13.7
Radon (Bq m^−3^) ^5^	104	96	127	142	144

^1^ Data available for the years 2002–2010; ^2^ PM_10_ = Particulate Matter 10 micrometers or less in diameter; Data available for the years 2006–2014; ^3^ PM_2.5_ = Particulate Matter 2.5 micrometers or less in diameter; Data available for the years 2011–2014; ^4^ Data available for the years 2010–2014; ^5^ Data available for the year 2013.

**Table 3 ijerph-16-00628-t003:** Comparison of cancer mortality trends in the monitored area (Rieti province) and the Latium region from 1982 to 2011.

Geographical Area	1982–1986	1987–1991	1992–1996	1997–2001	2002–2006	2007–2011	Mean 1982–2011
Rieti province	25.3	27.6	26.4	24.7	24.4	23.3	25.3
Latium region	31.4	32.3	30.7	29.2	28.2	27.5	29.9

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
