# Peer review of "Cancer Mortality Trend in Central Italy: Focus on A “Low Rate of Land Use” Area from 1982 to 2011"

_ijerph, 2019, doi:10.3390/ijerph16040628_

Round 1

Reviewer 1 Report

I think the article is very interesting and useful for comparing epidemiological studies in cancer in Italy and probably in different regions of Europe. However, I recommend corrections in the manuscript:
1. I suggest placing (in Table I or Table II) information on annual or monthly income per person in the compared regions, then: data on the population, education level, data on professional activity, and maybe some information on the use of alcohol or tobacco, smoking originating from cross-sectional studies (if any). We need to know if the population of the Rieti region is representative of the whole population in these aspects.
2. I expect the authors to discolor 3b and 3c in a similar way to Figure 3a

Author Response

We thank the reviewer for her/his constructive comments and suggestions on our manuscript.

I think the article is very interesting and useful for comparing epidemiological studies in cancer in Italy and probably in different regions of Europe. However, I recommend corrections in the manuscript:

1. I suggest placing (in Table I or Table II) information on annual or monthly income per person in the compared regions, then: data on the population, education level, data on professional activity, and maybe some information on the use of alcohol or tobacco, smoking originating from cross-sectional studies (if any). We need to know if the population of the Rieti region is representative of the whole population in these aspects.

A: Study area is Latium Region, that includes Rieti and other four provinces (Viterbo, Latina, Frosinone and Rome) in just 17.203 km² (5% of the whole Italian territory). Therefore, we compared cancer mortality data only among Latium inhabitants that do not differ each other for ethnicity, socio-economic characteristics and life-styles for the majority of the population. Unfortunately, detailed data routinely collected on these variables are available on regional and not on local/provincial basis. In order to clarify this point, we added the following sentence to paragraph 2.1: “In total, Latium Region presents a land area of 17203 km² (5.7% of the whole Italian territory).

2. I expect the authors to discolor 3b and 3c in a similar way to Figure 3a.

A: Ok, done.

A: Revised text was checked by a native speaker.

Reviewer 2 Report

The manuscript is well prepared and written. I recommend to accept this work for publication in IJERPH.

Please consider points listed below to improve the readiness.

Lines 77-78: "...of young persons (from 0 to 14 years old) x 100 at the years 1981, 1991, 2001, 2011 (Italian censuses)" change to "...of young persons (from 0 to 14 years old) x 100 at the years 1981, 1991, 2001 and 2011 (Italian censuses)"

Table1: "Frosinone Viterbo Italy" - change to bold.

Lines 93-94: "...low anthropogenic impact, low vehicular traffic [20], low municipal solid waste production [21]." change to "...low anthropogenic impact, low vehicular traffic [20] and low municipal solid waste production [21]."

Figure 4. The data could be presented on a single graph.

Lines 220-221:"However, rates for this cancer presented some gender differences: there was an increasing of 3.1% in males, while..." change to "However, rates for this cancer presented some gender differences: there was  3.1% increase in males, while..."

Author Response

We thank the reviewer for her/his constructive comments and suggestions on our manuscript.

The manuscript is well prepared and written. I recommend to accept this work for publication in IJERPH.

Please consider points listed below to improve the readiness.

Lines 77-78: "...of young persons (from 0 to 14 years old) x 100 at the years 1981, 1991, 2001, 2011 (Italian censuses)" change to "...of young persons (from 0 to 14 years old) x 100 at the years 1981, 1991, 2001 and 2011 (Italian censuses)"

A: Ok, done.

Table1: "Frosinone Viterbo Italy" - change to bold.

A: Ok, done.

Lines 93-94: "...low anthropogenic impact, low vehicular traffic [20], low municipal solid waste production [21]." change to "...low anthropogenic impact, low vehicular traffic [20] and low municipal solid waste production [21]."

A: Ok, done.

Figure 4. The data could be presented on a single graph.

A: Ok, done.

Lines 220-221:"However, rates for this cancer presented some gender differences: there was an increasing of 3.1% in males, while..." change to "However, rates for this cancer presented some gender differences: there was 3.1% increase in males, while..."

A: Ok, done.

A: Revised text was checked by a native speaker.

Reviewer 3 Report

Please find my comments in attached file

Author Response

The research under review addressed a very important health issue for millions of people around the world. It has been very well shaped structure and scientific establishment, with several related issue references.

The authors have established their research in international reports regarding both morbidity and mortality census data from cancer, and eight diagnosed pathogenic risk factors for its occurrence in the population. At the same time, they have created a minor human-geographic profile of cancer patients in Italy in general, and the study area in particular. Their research question clearly refers to two of these eight factors (environment and aging), as the area focusing on their study is characterized and delimited according to these two factors.

However, it is an undeniable fact that cancer is a multi-parametric disease and the study of the conditions that affect its appearance and effect on the population should not neglect any of the (at least) known factors that affect it. There is, therefore, a substantial dimension between the title of the study and its content.

A: We thank the reviewer for her/his comments and suggestions on our manuscript. We agree with the referee on the relevance of several risk factors linked to cancer; indeed, we specified that cancer is a multi-parametric disease in the second sentences of the Introduction section, as follows: “Over the decades, scientific evidences demonstrated the relevance of several risk factors, such as environmental pollutants [3], genomic alteration [4], infectious diseases [1], lifestyles and high body mass index, poor intake of fruit and vegetables, sedentarity, tobacco and alcohol consumption, aging [5].” We used data related to the five provinces of one of the 20 Italian regions. To better specify this point, we specified in the revised text that: “Study area is Latium Region, that includes Rieti and other four provinces (Viterbo, Latina, Frosinone and Rome) in just 17.203 km² (5% of the whole Italian territory).” Therefore, we compared cancer mortality data only among Latium inhabitants that do not differ each other for ethnicity, socio-economic characteristics and life-styles for the majority of the population. Unfortunately, detailed data routinely collected on these variables are available on regional and not on local/provincial basis.

An essential question that also arises here is the geographic context that the research focuses. That is, whether existing data at country level specifically showed the Rieti area as the only one to be considered. In other words, it not mentioned in the text such an approach, in order to delimit also other areas in Italy with the common characteristics of both the above mentioned factors (e.g. Cluster Analysis). Also, the risk between the supposed ones demarcates areas is not assessed in some way (e.g. Weighted Regression Analysis) in order to show actually that Rieti region is the most suitable for consideration.

If someone observes the matter from the geographic perspective, here we have to do with an undocumented choice of a particular area that ultimately leads to less objective conclusions.

And if response here the lack of data, at least initially the two previous methods of delimiting areas across the country and their calibration in terms of the risk of cancer deaths should be applied. Then the selection of the Rieti area (which would reasonably exist among them) would be obvious due to the fullness of the data it had. Otherwise, any conclusion drawn will be guided.

A: See the previous response to explain the choice to select the five province of Latium Region.

In relation to the methodology followed, what I think first of all is that the authors have correctly homogenised their data as they had to deal with two different revisions of the ICD.

On the other hand, they have used a ready-made software (Joinpoint 4.6.0.0) produced by the US National Cancer Institute. This is not necessarily prohibitive. Instead, it documents the good practices of expert on Cancer Institute, to have open software tool on international use from other specialists who need them.

But the question here is whether it is sufficient only that. According to the research’ results the authors proceed to interpretations, which really have their meaning, but their documentation is mainly based on charts produced either by Joinpoint 4.6.0.0 or by Excel, which they also used. Of course, in the discussion that follows, an attempt at documentation is made, with references to published national and international results of respective researches. Clearly aware of the lack of causality, the authors argue that “…The present study, performed with an ecological approach, cannot prove any causal correlation between an exposure and an event. Thus, it is not possible to give an explanation of the differences recovered between the studied provinces….”. However, research findings do not prohibit considering which of the disease agents involve in the study is the most aggravating. Simple application even a chi-square test, or a simple logistic regression, would be able to show some correlations between risk factors and patients or cancer deaths, so that the conclusions drawn are safer. Besides, I totally disagree with the authors’ claim, as studies conducted with an ecological approach use very much more tools and especially those of spatial statistical analysis.

A: We used the ready-made software Joinpoint 4.6.0.0 because, as highlighted also by the reviewer her/himself, “it documents the good practices of expert on Cancer Institute, to have open software tool on international use from other specialists who need them” and it we successfully used previously. Besides, we highlighted that the ecological approach is limited because it cannot be used to evaluate any causal relationship, but it is a useful study to provide a photograph of the spatial / temporal trend and, thus, we used it in order to provide a trend for cancer mortality for a very long period of time.

To conclude, I would like to make it clear to the good colleagues that on one hand, I recognized the effort of collecting and processing epidemiological data of 30 years and the time it takes for these projects. On the other hand, however, because this article deals with the effects of such a serious disease as cancer, which has afflicted, harassed and killed millions of people, researchers’ approaches must be thorough, very careful and well documented, so has to lead to scientific methods with the appropriate data processing, so that the scientific community can be guided by them and improve them.

For all the above-mentioned reasons I believe that this research should be considered as rejected.

A: We thank the reviewer for her/his comments and suggestions on our manuscript. It is undoubted that our paper presents some limitations, that we stated in the manuscript. On the other hand, as cancer afflicts, harasses and kills millions of people there is a need of epidemiological data in order to provide a picture and support strategic prevention decisions. For these reasons, we think that it could be important to make available a temporal trend of cancer mortality for a very long period (30 years) to the scientific community.

A: Revised text was checked by a native speaker.

Reviewer 4 Report

In this study, the authors estimated the cancer mortality trends from 1982 and 2011in a “low rate of land use” province of Latium Region (Rieti, central Italy) by using Joinpoint regression analysis. Authors showed a declining trend in mortality rates, both for males and females, over the time in the monitored area. In addition, their results emphasized that trachea-bronchus-lung cancer causes the greatest number of deaths followed by colorectal-anus, and stomach cancers. Furthermore, they compared the total mean of standardized mortality rate between Rieti province with Latium Region which showed that a decreasing temporal trend and a lower mortality for cancer in Rieti province when compared to the other Latium provinces. Based on these results, the authors claimed that the studied area could be considered as a “healthy” benchmark for studies in oncological diseases. Although the study is interesting and could be attributed to the implement of preventive initiatives of cancers, there are several points which need to be addressed:

1.      Since “low rate of land use” province of Latium Region (Rieti, central Italy) is characterized by a high prevalence of elder people, it is necessary to investigate if there are any changes in the trend of cancer mortality in those monitored areas after age adjustment as previous researchers reported ( e.g. PMID: 26091467).

2.      Authors mentioned that (line 148-150) “The analysis of standardized cancer mortality rates in relationship to the types of cancer highlighted that trachea-bronchus-lung cancer causes the greatest number of deaths in the monitored  area, followed by colorectal-anus, stomach, breast, prostate, liver, pancreas and leukemia.” ; however the authors did not  include any information/data about breast, prostate, liver, pancreas and leukemia. In addition, the authors mentioned that (line 212-213) “while females ones presented some peculiarities: breast cancer was the first one, followed by colorectal-anus and stomach” without any proof of data. Please included this information.

3.      Figure 3 explains only photographical trends of cancer mortality for the three main deadliest cancers in the monitored area from 1982 to 2011 in total, male, and female groups. For better statistical understanding, please represent this graph in table format and/or include all the necessary statistical parameters (e.g. APC, CI, p-value etc.) in the table/figure.

4.      Authors included annual mean concertation of few environmental pollutants in Latium provinces in table 2 and could not provide any casual correlation between exposures to those pollutants and cancer mortality trend in the monitored area which probably limits their hypothesis. Please make the necessary correction in conclusion section.

5.      In Figure 4, please include the mean standardized rate values on the top of each bar for better understanding.

6.      Authors could enrich their reference section by including more relevant articles that match with their current finding e.g. (PMID: 26045128).

Author Response

We thank the reviewer for her/his constructive comments and suggestions on our manuscript.

In this study, the authors estimated the cancer mortality trends from 1982 and 2011in a “low rate of land use” province of Latium Region (Rieti, central Italy) by using Joinpoint regression analysis. Authors showed a declining trend in mortality rates, both for males and females, over the time in the monitored area. In addition, their results emphasized that trachea-bronchus-lung cancer causes the greatest number of deaths followed by colorectal-anus, and stomach cancers. Furthermore, they compared the total mean of standardized mortality rate between Rieti province with Latium Region which showed that a decreasing temporal trend and a lower mortality for cancer in Rieti province when compared to the other Latium provinces. Based on these results, the authors claimed that the studied area could be considered as a “healthy” benchmark for studies in oncological diseases. Although the study is interesting and could be attributed to the implement of preventive initiatives of cancers, there are several points which need to be addressed:

1.      Since “low rate of land use” province of Latium Region (Rieti, central Italy) is characterized by a high prevalence of elder people, it is necessary to investigate if there are any changes in the trend of cancer mortality in those monitored areas after age adjustment as previous researchers reported ( e.g. PMID: 26091467).

A: Unfortunately, we have not data on cancer mortality grouped for age nor ages of death. Thus, we were not able to calculate cancer mortality rates in those monitored areas after age adjustment. We added this issue (and the related reference) in the limitations of the study, as follows: “These data were not grouped according to age nor reporting ages of death for each individual. Thus, we were not able to calculate cancer mortality in those monitored areas after age adjustment as elaborated by other researchers [40, 41].”. References are: 40. Politis, M., Higuera, G., Chang, L.R., Gomez, B., Bares, J., Motta, J. Trend analysis of cancer mortality and incidence in panama, using Joinpoint regression analysis. Medicine (Baltimore) 2015, 94, e970. DOI: 10.1097/MD.0000000000000970. 41. Rosso, T., Bertuccio, P., La Vecchia, C., Negri, E., Malvezzi, M. Cancer mortality trend analysis in Italy, 1980-2010, and predictions for 2015. Tumori 2015, 101, 664-75. DOI: 10.5301/tj.5000352.

2.      Authors mentioned that (line 148-150) “The analysis of standardized cancer mortality rates in relationship to the types of cancer highlighted that trachea-bronchus-lung cancer causes the greatest number of deaths in the monitored area, followed by colorectal-anus, stomach, breast, prostate, liver, pancreas and leukemia.” ; however the authors did not  include any information/data about breast, prostate, liver, pancreas and leukemia. In addition, the authors mentioned that (line 212-213) “while females’ ones presented some peculiarities: breast cancer was the first one, followed by colorectal-anus and stomach” without any proof of data. Please included this information.

A: A single paper on data obtained for all the studied cancers would have been a discussion too long. Thus, we focused our attention on the three deadliest cancers. We modified the text accordingly as follows: “Given this preliminary result, we focused the attention on the three deadliest cancers.”. We selected the three deadliest cancers for all population and, then, we evaluated possible differences between males and females. We did not report data on breast cancer and in the revised text we added that “data not shown”.

3.      Figure 3 explains only photographical trends of cancer mortality for the three main deadliest cancers in the monitored area from 1982 to 2011 in total, male, and female groups. For better statistical understanding, please represent this graph in table format and/or include all the necessary statistical parameters (e.g. APC, CI, p-value etc.) in the table/figure.

A: We evaluated the statistical significance of the differences in the trends by the use of Jointpoint regression analysis. The results evidenced that, despite a declining trend, no significant differences arose for the monitored period. Thus, we reported a more detailed picture of trends in figure 3, as we already showed the lack of a significant difference (Figure 2).

4.      Authors included annual mean concertation of few environmental pollutants in Latium provinces in table 2 and could not provide any casual correlation between exposures to those pollutants and cancer mortality trend in the monitored area which probably limits their hypothesis. Please make the necessary correction in conclusion section.

A: as suggested, we added the following sentence to the conclusion section: “Thus, even if we cannot provide any casual correlation between exposures to pollutants and cancer mortality trend, our results indicate that the studied area could be considered as a “healthy” benchmark for studies in oncological diseases.”.

5.      In Figure 4, please include the mean standardized rate values on the top of each bar for better understanding.

A: Ok, done.

6.      Authors could enrich their reference section by including more relevant articles that match with their current finding e.g. (PMID: 26045128).

A: Ok, done.

A: Revised text was checked by a native speaker.

Round 2

Reviewer 3 Report

The authors with the answers they gave to the revised edition have corrected some of the comments made at the first submission of the article.

There are, however, questions about:

1. The selection of the study area (despite the fact that the authors attempted to justify this choice based on the homogeneous characteristics of the population of the area) and

2. The interpretation of the results (despite the fact that the authors attempted to justify their work method by explaining that "the ecological approach is limited because it cannot be used to assess any causal relationship, but it is useful to study the creation of a photography of the spatial/temporal trend and so they used it to offer a tendency for cancer mortality for a very long time."

My disagreement with the above justifications is probably due to the fact that there is a different scientific origin through which the problem of cancer mortality is approached in this study.

The authors talk about a spatiotemporal approach to cancer mortality, but in my point of view, space does not play the important role it should be and therefore is not treated with the seriousness it should. The environmental factor under consideration (along with aging) has indeed a spatial dimension, which, in my opinion, is not sufficiently taken into account, as this research lacks the appropriate methodological tools that would achieve this goal.

On the other hand, however, I cannot ignore that: a. there is a systematic lack of research-related data for the entire Italian territory, which is a serious reason for limiting it to the area where the data is sufficient; b. there is no individual specialized in spatial analysis in the research group; and c. (perhaps the most important) the results may indeed be useful for further exploitation by the scientific community.

To conclude, I would have suggested to the honorable authors to reform their text in order to remove the term "spatial/temporal" as it is mentioned in it, so as to the results of this article are consistent with its methodology.

For those reasons, I believe that the article' s publishing can proceed with minor revisions.

Best regards,

Author Response

The authors with the answers they gave to the revised edition have corrected some of the comments made at the first submission of the article.

There are, however, questions about:

1. The selection of the study area (despite the fact that the authors attempted to justify this choice based on the homogeneous characteristics of the population of the area) and 2. The interpretation of the results (despite the fact that the authors attempted to justify their work method by explaining that "the ecological approach is limited because it cannot be used to assess any causal relationship, but it is useful to study the creation of a photography of the spatial/temporal trend and so they used it to offer a tendency for cancer mortality for a very long time." My disagreement with the above justifications is probably due to the fact that there is a different scientific origin through which the problem of cancer mortality is approached in this study. The authors talk about a spatiotemporal approach to cancer mortality, but in my point of view, space does not play the important role it should be and therefore is not treated with the seriousness it should. The environmental factor under consideration (along with aging) has indeed a spatial dimension, which, in my opinion, is not sufficiently taken into account, as this research lacks the appropriate methodological tools that would achieve this goal.

On the other hand, however, I cannot ignore that: a. there is a systematic lack of research-related data for the entire Italian territory, which is a serious reason for limiting it to the area where the data is sufficient; b. there is no individual specialized in spatial analysis in the research group; and c. (perhaps the most important) the results may indeed be useful for further exploitation by the scientific community.

To conclude, I would have suggested to the honorable authors to reform their text in order to remove the term "spatial/temporal" as it is mentioned in it, so as to the results of this article are consistent with its methodology.

For those reasons, I believe that the article' s publishing can proceed with minor revisions.

A: We thank the reviewer for her/his constructive comments and suggestions on our manuscript. We removed the terms “spatial” and “temporal” from the whole text.